# Enteral nutrition management in critically ill adult patients and its relationship with intensive care unit-acquired muscle weakness: A national cohort study

Ignacio Zaragoza-García[1,2]* , Susana Arias-Rivera[3], María Jesús Frade-Mera[1,4], Joan Daniel Martí[5], Elisabet Gallart[6], Alicia San José-Arribas[7], Tamara Raquel Velasco-Sanz[1,8], Eva Blazquez-Martínez[9], Marta Raurell-Torredà[10]

1 Department of Nursing, Faculty of Nursing, Physiotherapy and Podology, University Complutense of Madrid, Madrid, Spain, 2 Invecuid, Instituto de Investigación Sanitaria Hospital 12 de Octubre (imas12), Madrid, Spain, 3 University Hospital of Getafe, CIBER Enfermedades Respiratorias, Instituto de Salud Carlos III, Getafe, Spain, 4 Department of Critical Care, 12 Octubre University Hospital, Madrid, Spain, 5 Clinic University Hospital, Barcelona, Spain, 6 Department of Critical Care, Vall Hebron University Hospital, Barcelona, Spain, 7 Escola Universitaria d'Infermeria Sant Pau, Hospital de la Santa Creu i Sant Pau, Barcelona, Spain, 8 Department of Critical Care, San Carlos University Hospital, Madrid, Spain, 9 Bellvitge University Hospital, Hospitalet de Llobregat, Llobregat, Spain, 10 Department d'Infermeria Fonamental i medicoquirúrgica, Facultat d'Infermeria, Universitat de Barcelona, Barcelona, Spain

☯ These authors contributed equally to this work.
* izaragoz@ucm.es

**Data Availability Statement:** All relevant data are within the manuscript and its Supporting Information files.

## Abstract

### Objective

To assess the incidence and determinants of ICU-acquired muscle weakness (ICUAW) in adult patients with enteral nutrition (EN) during the first 7 days in the ICU and mechanical ventilation for at least 48 hours.

### Methods

A prospective, nationwide, multicentre cohort study in a national ICU network of 80 ICUs. ICU patients receiving invasive mechanical ventilation for at least 48 hours and EN the first 7 days of their ICU stay were included. The primary outcome was incidence of ICUAW. The secondary outcome was analysed, during days 3–7 of ICU stay, the relationship between demographic and clinical data to contribute to the onset of ICUAW, identify whether energy and protein intake can contribute independently to the onset of ICUAW and degree of compliance guidelines for EN.

### Results

319 patients were studied from 69 ICUs in our country. The incidence of ICUAW was 153/222 (68.9%; 95% CI [62.5%-74.7%]). Patients without ICUAW showed higher levels of active mobility (p = 0.018). The logistic regression analysis showed no effect on energy or protein intake on the onset of ICUAW. Overfeeding was observed on a significant proportion

**Funding:** This work was supported by 2018 european federation of critical care nursing associations (EfCCNa) Research Awards. The funders had no role in study design, data collection and analysis, decision to publish, or preparation of the manuscript.

**Competing interests:** The authors have declared that no competing interests exist.

of patient-days, while more overfeeding (as per US guidelines) was found among patients with obesity than those without (42.9% vs 12.5%; p<0.001). Protein intake was deficient (as per US/European guidelines) during ICU days 3–7.

## Conclusions

The incidence of ICUAW was high in this patient cohort. Early mobility was associated with a lower incidence of ICUAW. Significant overfeeding and deficient protein intake were observed. However, energy and protein intake alone were insufficient to explain ICUAW onset.

## Relevance to clinical practice

Low mobility, high incidence of ICUAW and low protein intake suggest the need to train, update and involve ICU professionals in nutritional care and the need for early mobilization of ICU patients.

## Introduction

Patients admitted to Intensive Care Units (ICUs) are subject to increased metabolic stress. Elevated catabolism requires nutritional resources for the body to perform anabolism adequately [1]. If oral intake is not possible, enteral nutrition (EN) is recommended over parenteral nutrition, because it has fewer complications [2]. Inappropriate management of enteral nutrition support in these patients can lead to malnutrition, a common finding in ICU patients [3], for which the incidence ranges from 39% to 50% of patients, depending on the country and ICU type [4].

Various authors have described possible causes of malnutrition in critically ill patients. Delayed initiation of nutrition support has been found in 60% of cases. In addition, an incorrect EN regimen can lead to under- or overfeeding, which, together with the inflammatory response typical for this metabolic state, can contribute to hyperglycaemia, loss of muscle mass and strength, prolonged rehabilitation, as well as an increase in comorbidities resulting in deteriorated quality of life in the long term [5].

Loss of muscle mass together with other factors, such as physical immobility, can lead to the onset of bilateral and symmetric neuromuscular complications, referred to as ICU-acquired muscle weakness (ICUAW), which contributes to significant functional impairment. Specifically, the muscles of the limbs and the diaphragm may become weak and atrophic, impairing patients' autonomy, prolonging mechanical ventilation, and increasing weaning time and length of hospital stays [6, 7].

The most studied predictors in ICUAW are related to gender, time on mechanical ventilation, length of ICU stay, age, more days on renal replacement therapy. On the other hand, the presence of delirium and being actively mobilised during the first 5 days in the ICU are considered protective factors [8, 9]. Some international bodies specialised in EN suggest the need for research on the relationship between EN and ICUAW, but due to lack of evidence, they do not yet make any recommendations in this regard [2, 10].

As a result, various international nutrition-related societies publish specific recommendations for critically ill patients. Recent studies suggest that diet-only interventions are insufficient to improve patients' nutritional status and reduce comorbidities, and this is now reflected in current recommendations [2]. To mitigate this deterioration, early mobilization in

the ICU is recommended [5]. The combination of nutrition plus exercise may modify the catabolic effects of critical illness, muscle wasting, and the development of ICUAW, which has been identified as a research priority [11].

Currently, no national multicentre studies have evaluated the management of EN in critically ill patients or the degree of mobility of these patients related to the incidence of ICUAW.

The aim of this study was to assess the incidence and determinants of ICUAW in adult patients with EN during the first 7 days in the ICU and receiving mechanical ventilation for at least 48 hours.

## Materials and methods

### Design

A prospective multicentre observational cohort study was conducted during four months (2019–2020) in a Spanish national ICU network of 80 ICUs.

### Data collection

Patients were recruited consecutively. The data were collected starting from day 3 of ICU admission. Inclusion criteria were adult patients receiving invasive mechanical ventilation (IMV) for at least 48 hours in an ICU and EN for at least the first 7 days of their ICU stay. Exclusion criteria were pregnant women, patients <18 years, those referred to the ICU from other hospitals, patients with primary neurologic or neuromuscular pathology, those unable to walk, recent limb amputees, users of orthopaedic devices and patients with body mass index (BMI) >35.

### Sample/Participants

The minimum sample size was 316, calculated according to the 46% incidence of ICUAW found in a sample of 1421 patients by Stevens et al. [12], a confidence level of 95%, an estimated standard error of 5 and an expected loss of 5%.

### Ethical considerations

The study was approved by the Ethics and Clinical Research Committees of the participating sites under reference protocol PI16/00771. Written informed consent was obtained. The relevant STROBE checklist was followed for reporting the study.

### Research variables and measures

**Primary outcome.** The primary outcome was incidence of ICUAW, assessed by the Medical Research Council Scale (MRC-Sum score) following the assessment protocol described by Hermans [13]. ICUAW was diagnosed for values lower than 48 out of 60 (the maximum score) in the first measure of MRC (baseline MRC) [14].

The measure of MRC was conducted after the first awakening of the patient, with the patient fully awake. See S1 File. Measurement tools.

**Secondary outcomes.** The secondary outcome were, on the one hand, analysis of the relationship between demographic and clinical data contributing to the onset of ICUAW during days 3–7 of ICU stay. On the other hand, we proceeded to identify whether energy or protein intake during 3–7 days of the ICU stay, taking into account the US and European recommendation, can contribute independently to the onset of ICUAW. Finally, the degree of compliance with current US and European guidelines for target dietary intake in EN during the acute phase (days 3–7) of ICU admission was analysed.

Specifically, the following recommendations were used for reference in the study [2, 15]:

Target energy and protein intake: According to ASPEN (American Society for Parenteral and Enteral Nutrition) guidelines [15]: target energy and protein intake should be 25–30 kcal/kg/day and 1.2–2 g/kg/day, respectively. During the first week, trophic EN is permitted. For patients with BMI $\geq$30 kg/m$^2$ the energy target is 11–14 kcal/kg/day actual body weight/day and the protein target is 2 g/kg ideal body weight/day.

According to ESPEN (European Society for Clinical Nutrition and Metabolism) guidelines [2]: target energy and protein intake is 20–25 kcal/kg/day and 1.3 g/kg/day delivered progressively, respectively. During the first week, trophic EN is permitted. Actual body weight is used for patients with BMI $\leq$25 kg/m$^2$ and adjusted body weight for BMI >25 kg/m$^2$.

Other recommendations discussed were interruptions to EN should be avoided. It is recommended that stopping feeding to evaluate oral tolerance should be limited to once daily at the most. In addition, gastric residual volume < 500 mL indicates EN tolerance. Finally, insulin therapy should be used to control blood glucose levels and blood glucose levels should be maintained at <180 mg/dL.

**Variables.** Independent variables related to the patient's baseline condition as well as hospital admission variables were collected. Specifically, age, gender, and BMI, diagnosis on admission, Barthel and Charlson index, and APACHE II scores were collected. All parameters were collected from the medical records by a collaborating research nurse. See supplementary material for definitions and classifications.

The principal dependent variable that was collected was presence of ICUAW according to MRC sum-score, conducted by a physiotherapist. Secondary dependent variables were energy and protein intake via EN, level of mobility, continuous renal replacement therapy (CRRT), airway management, ICUAW-related drugs, vasopressors, moderate and severe hyperglycaemia.

All these variables were collected during days 3–7 of ICU stay: ICU Mobility Scale (IMS) score. IMS is a 10-point scale ranges from 0 (patient immobile lying in bed) to 10 (independent ambulation). The IMS was categorized using a binary system (where <4 represents, in-bed activities, and $\geq$4, active out-of-bed mobilization); Days on which the patient requires CRRT; Type of airway management (invasive mechanical ventilation (IMV) or no IMV); ICUAW-related drugs, understood as cumulative doses of drugs such as neuromuscular blocking agents, steroids [methylprednisolone, dexamethasone, and hydrocortisone in mg equivalent dose] and aminoglycosides; Administered doses of Vasopressors (epinephrine, adrenaline, noradrenaline, dopamine and dobutamine). In both cases above intravenous administration is considered (continuous infusion, stat dose, and bolus injection on demand); Moderate (glycaemia >181 and $\leq$215 mg/dl) or severe hyperglycaemia ($\geq$216 mg/dl) of the total blood glucose results on day 3–7 of the ICU stay multiplied by 100.

A team of trained professionals recorded the variables. Detailed of the measurement tools are provided in the S1 File.

**Data analysis.** Categorical variables were expressed as frequency and percentage, using Fisher or Chi-squared test for comparison between groups. Quantitative variables were expressed as mean and standard deviation (SD) or median and interquartile range (IQR), and groups were compared using Student-t or Mann–Whitney U test. To study the correlation between quantitative variables (actual body weight, energy and protein intake), Pearson or Spearman was used. A multivariate analysis was used to investigate the association between EN during 3–7 days of the ICU stay (energy and protein administration, days with overfeeding and days with protein >0.8 g/kg/day) and ICUAW, also controlling other explanatory variables: baseline variables (age, gender, BMI, Barthel and Charlson scores) and those related to ICUAW onset during days 3–7 of the ICU stay (days with CCRT, doses of ICUAW-related

drugs and vasopressors, days with IMS ≥4, and days with moderate and severe hyperglycaemia). Data were analysed using SPSS 25.0.

## Results

We analysed 319 patients, corresponding to 1595 EN days and 69 ICUs in our country (Fig 1).

The incidence of ICUAW was 68.9% (153/222 patients; 95% CI [62.5%-74.7%]). In 30.4% (97/319 patients; 95% CI [25.6%-35.7%]), the MRC assessment was unfeasible. Among the patients with ICUAW, females were at higher risk than males and the most prevalent diagnosis was sepsis. Patients with ICUAW had higher rates of comorbidity (Charlson), were more dependent (Barthel) and had greater disease severity (APACHE), but these results were not statistically significant (Table 1). More overweight and, conversely, fewer obese patients developed ICUAW (p<0.05 in both cases) (Table 1).

On ICU days 3–7, although the general cohort had low active mobility out of bed (IMS≥4), ICUAW patients had significantly lower values during this period (p<0.001) (Table 2). In addition, ICUAW patients received significantly more vasopressors (p = 0.029) and had more days of IMV (p = 0.032). No significant differences were found in median days of CRRT, of severe or moderate hyperglycaemia, or of administration of ICUAW-related drugs (Table 2) in the same period. Of the patients who developed ICUAW, 67.4% (273/405 patient/days) had deep sedation (RASS-3-5) vs 47.7% (83/174 patient/days) of those who did not have ICUAW. Patients in whom ICUAW could not be assessed had significantly more days of deep sedation than those in whom ICUAW could be assessed (87.3% (261/299) vs 61.5% (356/579); p<0.001).

Energy intake during days 3–7 was similar among patients who did and did not develop ICUAW, independently of which guidelines were followed (ASPEN or ESPEN). Likewise, no differences were observed in the percentage of patients with overfeeding or number of days of overfeeding when comparing patients with and without ICUAW (Table 3). Patients receiving propofol had a median energy intake of 188.0 kcal/day [62.7–380.6 kcal/day] over a total of 727 patient/days.

With regard to protein intake during days 3–7 in all groups, independently of which guidelines were followed (ASPEN or ESPEN), no differences were observed between number of days with >0.8 g/kg/day and ICUAW onset (Table 3).

Median protein intake was below 0.8 g/kg/day, and the median in obese patients with ICUAW (as per ASPEN guidelines) was closer to the recommended value, at 0.77 g/kg/day [0.48–0.99] (Table 3).

The logistic regression analysis for ICU days 3–7 showed no effect of energy or protein intake on the onset of ICUAW. Neither could ICUAW be explained by the increase in days with overfeeding (S2 File). The days with overfeeding on ICU days 3–7 showed an OR of 1.085 [0.934–1.261]; p = 0.286. This remained after adjusting for baseline variables (OR: 1.109 [0.948–1.296]; p = 0.197). The results were similar after adjusting for ICU stay variables (OR: 1.106 [0.945–1.294]; 0.209) and after adjusting for all variables (baseline and ICU stay variables) (OR:1.128 [0.956–1.332]; p = 0.154).

Median daily energy intake was close to recommended levels during the first week of the ICU stay, except for obese patients, who were found to receive slightly above the recommended energy intake levels according to the US guidelines (S1 Fig).

The degree of compliance with energy intake depends on which recommendations are considered. Overfeeding was observed according to US and European guidelines. Using the US guidelines and considering patients with BMI <30kg/m$^2$, overfeeding was found on 12.5% patients/day; 95% CI [10.8%-14.5%] whereas for patients with BMI ≥30kg/m$^2$ the rate was

Figure 1. Flow diagram showing patients' movement through the study

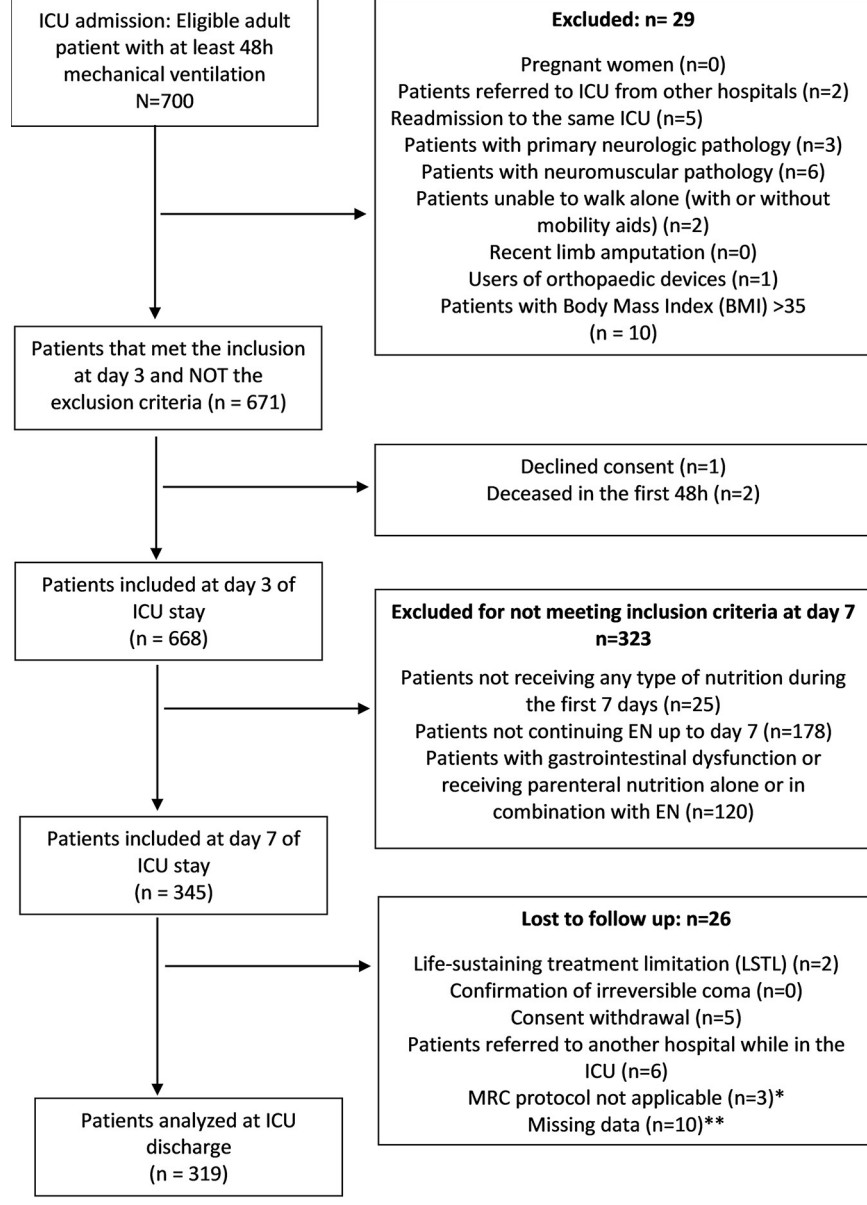

**Fig 1. Flow diagram showing patients' movement through the study.**

42.9% patients/day; 95% CI [38.0%-48.0%] and according to the European guidelines the rate was 43.1% patients/day; 95% CI [40.7%-45.5%] (S1 Table).

Median protein administration was low throughout days 3–7 of the ICU stay according to US and European guidelines (S1 Fig and S1 Table). A third of patients received less than 0.5 g/kg/day of protein during the first week (S1 Table).

A high percentage of patient-days showed glycaemia <180mg/dl. Patients without ICUAW had a significantly higher proportion of patient-days with glycaemia <180 mg/dl (p = 0.006)

**Table 1. General characteristics of the study population.**

| | | Patients with EN who developed ICUAW n = 153 (48.0%) | Patients with EN who did NOT develop ICUAW n = 69 (21.6%) | p value |
|---|---|---|---|---|
| Gender | Female | 52 (34.0%) | 14 (20.3%) | 0.041 |
| Age, years | | 68.0 [55.0–76.0] | 63.0 [47.5–74.5] | 0.079 |
| Dx. on Admission | Sepsis | 32 (20.9%) | 9 (13.0%) | <0.001 |
| | Trauma | 10 (6.5%) | 7 (10.1%) | 0.467 |
| | Neurosurgery | 4 (2.6%) | 0 (0%) | - |
| | Cardiovascular surgery | 15 (9.8%) | 4 (5.8%) | 0.012 |
| | Other surgeries | 18 (11.8%) | 8 (11.6%) | 0.050 |
| | Overdose | 3 (2.0%) | 2 (2.9%) | 0.655 |
| | Other medical patients | 71 (46.4%) | 39 (56.5%) | 0.002 |
| BMI (kg/m$^2$) | | 27.1 [24.3–30.3] | 26.7 [24.0–30.8] | 0.752 |
| BMI | Underweight | 2 (1.3%) | 0 (0.0%) | - |
| | Normal | 46 (30.1%) | 23 (33.3%) | 0.006 |
| | Overweight | 66 (43.1%) | 24 (34.8%) | <0.001 |
| | Obese | 39 (25.5%) | 22 (31.9%) | 0.030 |
| Barthel | | 100 [95–100] | 100 [95–100] | 0.933 |
| Charlson index | | 5.0 [2.0–7.0] | 4.0 [1.0–6.0] | 0.247 |
| APACHE II[a] | | 23 [18–28] | 21 [16–27] | 0.537 |

EN: enteral nutrition; ICUAW: intensive care unit-acquired muscle weakness; n: sample; %: percentage; BMI: body mass index; MRC: Medical Research Council scale.
[a]APACHE II (assessed in 45 patients without ICUAW, 67 with ICUAW and 59 with missing ICUAW data). Categorical variables are expressed as frequency and percentage (n (%)) and quantitative variables with non-normal distribution as median [25th -75th percentile]

(S2 Table). On most patient/days there was one or zero interruptions or pauses in EN. Gastric residual volume (GRV) was <500 ml on most patient-days. No differences were found between patients with or without ICUAW for glycaemia or GRV (S2 Table).

A weak correlation was found between patients' actual body weight and energy (kcal/kg/day) (r = -0.121; p<0.031) and proteins (g/kg/day) (r = -0.112; p<0.045) delivery.

**Table 2. ICU variables, by ICUAW onset on days 3–7 of ICU stay.**

| | Patients with EN who developed ICUAW n = 153 (48.0%) | Patients with EN who did NOT develop ICUAW n = 69 (21.6%) | p value |
|---|---|---|---|
| Days with IMS≥4 | 0.0 [0.0–0.0] / 0.02±0.14 | 0.0 [0.0–0.0] / 0.12±0.44 | <0.001 |
| Days with CRRT | 0.0 [0–0] | 0.0 [0–0] | 0.327 |
| Days according to Airway management | | | |
| • IMV | 5.0 [5.0–5.0] | 5.0 [4.0–5.0] | 0.032 |
| • no IMV | 0.0 [0.0–0.0] | 0.0 [0.0–1.0] | 0.028 |
| ICUAW-related drugs (mg) | 60.0 [0.0–307.4] | 0.0 [0.0–240.0] | 0.111 |
| Vasopressors (mg) | 24.2 [0.0–123.5] | 7.7 [0.0–39.2] | 0.029 |
| Moderate hyperglycaemia (rate) | 5.6 [0.0–20.9] | 5.9 [0.0–17.7] | 0.386 |
| Severe hyperglycaemia (rate) | 0.0 [0.0–18.0] | 0.0 [0.0–11.5] | 0.223 |

EN: enteral nutrition; ICUAW: intensive care unit-acquired muscle weakness; n: sample; %: percentage; IMV: invasive mechanical ventilation; IMS: ICU mobility scale; CRRT: continuous renal replacement therapy; SD: standard deviation. Quantitative variables are expressed as median [25th -75th percentile] or mean and SD.

Table 3. Energy and protein intake via EN and association with ICUAW on ICU days 3–7.

| | Patients with EN who developed ICUAW n = 153 (48.0%) | Patients with EN who did NOT develop ICUAW n = 69 (21.6%) | p value |
|---|---|---|---|
| **Energy. 2016 ASPEN guidelines** | | | |
| Energy (BMI <30 kg/m$^2$) | 16.8 [10.6–22.4] | 16.2 [11.1–22.5] | 0.899 |
| Overfeeding (BMI <30 kg/m$^2$)n (%) | 10 (8.8%) | 5 (10.6%) | 0.555 |
| Days with overfeeding (BMI <30 kg/m$^2$) | 0.0 / [0.0–0.0] | 0.0 / [0.0–0.0] | 0.998 |
| Energy (BMI >30 kg/m$^2$) | 12.6 [9.1–18.3] | 12.0 [9.7–17.2] | 0.988 |
| Overfeeding (BMI >30 kg/m$^2$)n (%) | 12 (30.8%) | 8 (36.4%) | 0.778 |
| Days with overfeeding (BMI >30 kg/m$^2$) | 2.0 [0.0–4.0] | 1.5 [0.0–3.0] | 0.787 |
| **Energy. 2019 ESPEN guidelines** | | | |
| Energy | 16.7 [10.7–23.0] | 15.9 [12.0–22.4] | 0.916 |
| Overfeeding n (%) | 62 (40.5%) | 26 (37.7%) | 0.767 |
| Days with overfeeding | 2 [0–4] | 1 [0–3] | 0.330 |
| **Protein. 2016 ASPEN guidelines** | | | |
| Prot (BMI <30 kg/m2) | 0.65 [0.46–0.89] | 0.69 [0.44–0.91] | 0.873 |
| Days with protein >0.8 g/kg/day (BMI <30 kg/m2) | 1.0 [0.0–3.3] | 2.0 [0.0–4.0] | 0.598 |
| Protein (BMI >30 kg/m2) | 0.77 [0.48–0.99] | 0.69 [0.52–0.87] | 0.409 |
| Days with protein >0.8 g/kg/day (BMI >30 kg/m2) | 3.0 [0.0–4.0] | 1.0 [0.0–3.0] | 0.238 |
| **Protein. 2019 ESPEN guidelines** | | | |
| Protein | 0.69 [0.46–0.92] | 0.68 [0.46–0.91] | 0.793 |
| Days with protein >0.8 g/kg/day | 2 [0–4] | 2 [0–4] | 0.654 |

EN: enteral nutrition; ICUAW: intensive care unit-acquired muscle weakness; n: sample; %: percentage. Energy is calculated as Kcal/Kg/day; protein as g/Kg/day.
Categorical variables are expressed as frequency and percentage (n (%)) and quantitative variables as median [25th -75th percentile].

## Discussion

The 68.9% incidence of ICUAW found in this study was higher than the 40% incidence (95% CI [38–42]) reported in a systematic review [16]. However, the percentage of patients without MRC assessment during the ICU stay was similar (26% IC 95% [16, 24–28]). Missing ICUAW data is explained by the patients in whom it was impossible to perform MRC due to insufficient awakening and comprehension (97/97 [100%] patients), which in itself is considered a factor that hinders early mobilization [9, 16, 17].

As in other studies, ICUAW was found predominantly in females and patients with sepsis [18]. We found an association between overweight and ICUAW onset, although the opposite occurred in the case of obesity. A study conducted in obese and non-obese septic mice [19] found that sepsis reduced body weight similarly in both groups, but there was attenuated muscle wasting and weakness in the obese mice. This is known as the 'obesity paradox'.

Mechanical ventilation can lead to a daily muscle loss of 1–2% [12]. In addition, the side effects of inappropriate nutrition support include hyperglycaemia, muscle loss, prolonged weaning from MV, and delayed rehabilitation [1, 5].

A review suggests that EN support alone is insufficient to reduce early muscle catabolism, proposing a combination of early mobilization and optimal rehabilitation [20]. A current study, investigated the association between these variables, finding that high protein intake and early mobilization preserves muscle mass [21]. Similarly, other study conducted a clinical trial with three groups (early mobilization, early mobilization and enteral nutrition protocol based on ESPEN guidelines, and control group), and found an improvement in ICUAW in

both intervention groups versus the control group [10]. Despite this, there was little difference between the intervention groups, except for the improvement in muscle strength found in the enteral nutrition and early mobilization group versus only early mobilization. The patients in our study had deficient protein intake and few achieved early active mobility. According to Hermans [22], higher levels of patient mobilization are achieved when physiotherapists lead mobilization decisions. Our study found that despite the low active mobility on days 3–7, low mobility is associated with the onset of ICUAW (p = 0.018).

We found no differences in drug administration (neuromuscular blocking agents, steroids and aminoglycosides) with regard to ICUAW, as corroborated by other recent study [9] except for the administration of vasopressors, which was also described by other authors [23]. Our study population was found a high percentage of overfeeding, but insufficient protein intake. Despite there being some controversy, some authors suggest that protein deficiency may lead to muscle deterioration and risk for ICUAW [21, 24], yet we found no differences in protein intake between patients who developed ICUAW or not. This finding may, however, be due to generalized low protein delivery [25]. In our case, a third of patient/days were below 0.5 g protein/kg/day, which is defined in the European guidelines as a low protein diet. Therefore, according to our results, the onset of ICUAW appears to be unrelated to protein intake, because although protein intake was low in most patients, some of those patients did not develop ICUAW.

## ICUAW-related guideline recommendations: Energy and protein administration

A high percentage of patients received trophic EN or were below 80% of the US recommendations [15] for target energy during the first week. However, current evidence and the European recommendations [2], along with the most recent US guideline update [26], show a tendency towards lower energy intake during this period. Arabi et al. [27], noted that anorexia is a common characteristic of critically ill patients. However, during the acute phase, full nutrient provision can be detrimental because it inhibits autophagy, giving cause for concern considering that in our study, overfeeding, defined as "energy administration of 110% above the defined target" [2], was found on almost half of patient/days (43.1%), applying the European recommendations, and in 42.9% of obese patients, applying the 2016 US recommendations. Despite this, in our study we have not been able to establish overfeeding as defined in the European guidelines as an explanation for ICUAW.

Although some authors question the optimal amount of protein to deliver, most agree that early initiation is more important than energy provision [28].

According to both guidelines, protein intake was insufficient in most patients in our sample. Cahill et al. [29] audited 20 countries to evaluate protein support and concluded that only 2.5% of hospitals achieved >80% of the protein target. Similarly, more recent studies have found below-target protein intake, specifically 52% (±30%) of the prescribed goal [30] and 10–12% of the total calorie intake, instead of 24–32% [31]. Furthermore, although our study had few patients with CRRT patient/days, ongoing use of these therapies may reduce the protein available for muscle formation [15, 32], and this would worsen protein intake deficiency.

Several studies have described various barriers to delivering the nutritional target in critically ill patients. A study identified three factors involved in compliance, which are related to patient, clinical, and site-specific considerations [30]. A Canadian study reported on a nutritional improvement programme in the ICU whereby patients attained over 80% of recommended target energy and protein intakes [33]. This success was attributed to 1) Presence of registered dietitians in the ICU; 2) Education of the clinical team regarding the need for good

nutritional practice; 3) Encouragement of a culture of interest in bedside nutritional care among all ICU staff. Furthermore, in our results, a weak inverse correlation between weight and kcal/kg/day and proteins/kg/day may suggest that EN was administered through a standard regimen, regardless of weight or patient state. Similar results were reported in a study conducted in 46 countries over 7 years, observing that patients were undernourished because EN was not guided by weight or disease status [34].

Furthermore, Peterson et al. [35] found that no enteral product is able to provide adequate protein intake without excess calorie intake. This observation is important because the current trend is for permissive underfeeding [36]. McCall et al. [33] reported that they increased protein intake by delivering additional protein in powder boluses. Other authors have proposed the use of parenteral amino acids [37], although this route of administration appears to result in lower protein availability (83% vs complete protein) [38].

## Other recommendations related to ICUAW prevention

Patients with ICUAW had fewer patient-days with glycaemia <180 mg/dl. Although hyperglycaemia was not found to be a risk factor for developing ICUAW in this study, other authors have found an independent relationship between ICUAW and more than 3 days of hyperglycaemia [22, 39].

EN cessation was observed on a third of patient/days, which contrasts with few patient/days with GRV >500 ml and a high energy vs poor protein intake. Unlike other authors' findings [40], this study appears to show that EN cessations are not the main cause of inappropriate nutrition support.

In view of various authors' findings and ours, it seems reasonable to combine various actions, including the use of up-to-date EN protocols in all ICUs and the presence in ICUs of professionals trained in critical care nutrition, and with ICU care team members trained and motivated to provide early mobilization, who can monitor patients throughout their stay and be involved in discharge plans [6, 41–43].

## Limitations

The lack of a cohort of patients attaining protein goals limits the results on a potential association with ICUAW. Measuring ICUAW by means of the MRC scale requires patients' cooperation, which may have caused a delay or absence in diagnosing ICUAW. The patient's actual weight was only recorded on admission and at no other time during the patient's stay. Like other authors, we found a lack of reliable instruments available in the ICUs to measure body weight. In addition, weight estimation is hard because of fluid loss and gain, and changes in lean tissue mass [44]. Patients' actual energy requirements could not be measured due to a generalised absence of indirect calorimetry techniques in the ICUs. Instead, energy requirements were estimated following the general recommendations in international guidelines. Finally, use of parenteral nutrition alone or in combination with EN was not investigated, and some authors have found that parenteral nutrition is detrimental in ICUAW prevention [14, 18].

## Implications and recommendations for practice

This study highlights the need for better adherence to international enteral feeding guidelines among patients admitted to the ICU. The existence of low mobility, high incidence of ICUAW and low protein intake suggest a need to continue future research to further inform the nutrition–early mobilization binomial, which has recently been observed for the first time. Such an approach–considering nutrition as a priority but never alone–will enable us to overcome the

undesirable effects of ICUAW. We believe that it is necessary to train, update and involve ICU professionals in nutritional care and the need for early mobilization of ICU patients.

## Conclusions

The incidence of ICUAW was high in patients receiving EN for at least one week.

Early mobilization is associated with lower incidence of ICUAW. Energy and protein intake alone was insufficient to explain the onset of ICUAW. The influence of protein intake on ICUAW was unclear, because significant protein deficiency was found in almost all patients throughout days 3–7 of the ICU stay. Although overfeeding was a common finding in this patient population, we were unable to confirm an association between overfeeding and ICUAW onset. Despite adequate compliance with some recommendations, a high percentage of patients were malnourished according to the guidelines.

Future studies are needed in which early mobilization is more widely implemented and nutritional requirements are calculated according to individual patients' baseline situation and clinical condition, thereby permitting further investigation of the onset of ICUAW in critically ill patients.

## Supporting information

**S1 File. Measurement tools.**
(PDF)

**S2 File. Logistic regression.**
(PDF)

**S1 Fig. Daily energy and protein intake via enteral nutrition, measured by kg of body weight and day.**
(TIF)

**S1 Table. Energy and protein intake by target recommendation.**
(PDF)

**S2 Table. Other recommendations for enteral nutrition on ICU days 3 to 7.**
(PDF)

## Acknowledgments

To Sociedad Española de Enfermería Intensiva y Unidades Coronarias (SEEIUC) who promoted the study.

To the MoviPre Group who collaborated in the data collection.

## Author Contributions

**Conceptualization:** Ignacio Zaragoza-García, Susana Arias-Rivera, María Jesús Frade-Mera, Joan Daniel Martí, Elisabet Gallart, Alicia San José-Arribas, Tamara Raquel Velasco-Sanz, Eva Blazquez-Martínez, Marta Raurell-Torredà.

**Data curation:** Ignacio Zaragoza-García, Susana Arias-Rivera, María Jesús Frade-Mera, Joan Daniel Martí, Elisabet Gallart, Alicia San José-Arribas, Tamara Raquel Velasco-Sanz, Eva Blazquez-Martínez, Marta Raurell-Torredà.

**Formal analysis:** Ignacio Zaragoza-García, Susana Arias-Rivera, María Jesús Frade-Mera, Alicia San José-Arribas, Tamara Raquel Velasco-Sanz, Eva Blazquez-Martínez, Marta Raurell-Torredà.

**Funding acquisition:** Alicia San José-Arribas, Marta Raurell-Torredà.

**Investigation:** Ignacio Zaragoza-García, Susana Arias-Rivera, María Jesús Frade-Mera, Alicia San José-Arribas.

**Methodology:** Ignacio Zaragoza-García, María Jesús Frade-Mera, Joan Daniel Martí, Elisabet Gallart, Eva Blazquez-Martínez, Marta Raurell-Torredà.

**Supervision:** Susana Arias-Rivera, María Jesús Frade-Mera, Elisabet Gallart, Alicia San José-Arribas, Marta Raurell-Torredà.

**Writing – original draft:** Ignacio Zaragoza-García, Marta Raurell-Torredà.

**Writing – review & editing:** Ignacio Zaragoza-García, Susana Arias-Rivera, María Jesús Frade-Mera, Joan Daniel Martí, Elisabet Gallart, Alicia San José-Arribas, Tamara Raquel Velasco-Sanz, Eva Blazquez-Martínez, Marta Raurell-Torredà.

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
