## [Decision Letter · Decision Letter 0]

10 Apr 2023

PONE-D-23-04050Enteral nutrition management in critically ill adult patients and its relationship with intensive care unit-acquired muscle weakness: a national cohort studyPLOS ONE

Dear Dr. Zaragoza-García,

Thank you for submitting your manuscript to PLOS ONE. After careful consideration, we feel that it has merit but does not fully meet PLOS ONE’s publication criteria as it currently stands. Therefore, we invite you to submit a revised version of the manuscript that addresses the points raised during the review process.

We look forward to receiving your revised manuscript.

Kind regards,

Sebastien Kenmoe

Academic Editor

PLOS ONE

“This work was supported by 2018 european federation of critical care nursing associations (EfCCNa) Research Awards.”

Reviewers' comments:

Reviewer's Responses to Questions

**Comments to the Author**

1. Is the manuscript technically sound, and do the data support the conclusions?

Reviewer #1: Yes

Reviewer #2: Partly

2. Has the statistical analysis been performed appropriately and rigorously? 

Reviewer #1: Yes

Reviewer #2: N/A

3. Have the authors made all data underlying the findings in their manuscript fully available?

Reviewer #1: Yes

Reviewer #2: Yes

4. Is the manuscript presented in an intelligible fashion and written in standard English?

Reviewer #1: Yes

Reviewer #2: Yes

5. Review Comments to the Author

Reviewer #1: Reword the main objective so that it has a single action verb. For example, assess the incidence and determinants of ICUAW....

In abstract, methodology to be improved

Methods

Specify year, country in methodology

Specify demographic and clinical parameters, and how they were obtained

Reviewer #2: Important clinical issue are discussed in the study under revision. I pointed out have some work aspects to improve the manuscript.

Major League:

In the introduction, the different ICUAW predictors are not sufficiently explained in order to reinforce the orientation of the problem. Explain the reasons for this choice of predictors based on evidence data.

In the discussion section, the incidence reported does not reflect the reality of the study (48% vs 68.9%), which constitutes a bias in the interpretation of the results and the different comparisons. The entire study population was not assessed on this criterion.

Minor comments:

line 104: EN is already defined in line 97.

Lines 173-220: Write this section in simple, understandable text instead of listing items.

Discussion: Most of the information can be given by just adopting the numbered reference since the names of the authors appear repeatedly. Review the writing style at this level.

6. PLOS authors have the option to publish the peer review history of their article (what does this mean?). If published, this will include your full peer review and any attached files.

Reviewer #1: **Yes: **Guy Roussel Takuissu Nguemto

Reviewer #2: No

---

## [Author Response · Author response to Decision Letter 0]

15 May 2023

Reviewers' comments:

Reviewer's Responses to Questions

Comments to the Author

5. Review Comments to the Author

-Reviewer #1: Reword the main objective so that it has a single action verb. For example, assess the incidence and determinants of ICUAW....

We agree with your comment. We have made the modification to the objective and it now has only one action verb. Modifications have been made in the summary (line 47) and in the general text (line 127). 

-In abstract, methodology to be improved

We agree with your suggestion. We have completed the methodology section in the abstract (lines 50-52).

-Methods

Specify year, country in methodology

We have added the year and country in methodology (line 142).

-Specify demographic and clinical parameters, and how they were obtained

We have completed the variables section (in the methodology section), specifying the demographic and clinical parameters, as well as how they were obtained (lines 195-204). At the end of this section, it is detailed that the professionals who collected data were previously trained (lines 218-219).

-Reviewer #2: Important clinical issue are discussed in the study under revision. I pointed out have some work aspects to improve the manuscript.

-Major League:

In the introduction, the different ICUAW predictors are not sufficiently explained in order to reinforce the orientation of the problem. Explain the reasons for this choice of predictors based on evidence data.

We have reviewed the section of the introduction where factors related to ICUAW are considered. We have added a description of these factors in relation to the available evidence (lines 110-115).

-In the discussion section, the incidence reported does not reflect the reality of the study (48% vs 68.9%), which constitutes a bias in the interpretation of the results and the different comparisons. The entire study population was not assessed on this criterion.

It is true that some of the patients collected could not obtain the ICUAW assessment because of the need for the patient to be awake in order to use the MRC scale (this limitation has been described in the limitations section). 

Other authors who use the same tool have found the same problem. (Raurell-Torredá 2021; Eggmann 2020). In the same way, these results are in line with the systematic review by Appleton (Appleton 2015): “Approximately a quarter of patients were not able to comply with clinical evaluation and this may be responsible for potential underreporting of this condition. Patients unable to comply with clinical assessment tend to have a higher mortality rate and potentially have greater encephalopathy, both of which are associated with increased incidences of ICUAW”.

As pointed out in the Appleton’s systematic review (2015), the incidence of ICUAW is likely to be higher than that obtained by clinical assessment (such as MRC), is likely to be explained in part by higher rates of incomplete testing. The incidence of ICUAW is lower when it is diagnosed with a clinical assessment (413/1276, 32%, 95% CI 30–35%) than when it is diagnosed using electrophysiological techniques (749/1591, 47%, 95% CI 45–50%). As such, for the cases where the MRC assessment could not be conducted, it is likely that these patients would have been diagnosed with ICUAW if other techniques had been used, such as ultrasonography or electrophysiology. Furthermore, due to the lack of a formal training program or a standardized protocol to educate clinicians in this technique, an adequate training of the personnel in the 80 ICUs participants of the study was not feasible.

Additionally, it is worth mentioning that the MRC scale is the technique for the diagnosis of ICUAW recommended by the consensus of experts (Needham 2017).

The calculation of the incidence of ICUAW was obtained with those individuals for whom MRC could be measured. 

Furthermore, it is important to note that if we use the whole population to calculate ICUAW, including those who could not be measured for MRC, we could have an erroneous result, as the missing patients have been considered as non-ICUAW (153/319=0.48 (48%)). This is contrary to Appelton's systematic review.

On the other hand, we have decided not to provide information from the general population so that readers would not find the tables confusing. However, if you find this information useful, we can add it so that you can see the results for the whole population. All tables are added below with an additional column providing the results for the whole population.

References:

Raurell-Torredà M, Arias-Rivera S, Martí JD, et al. Care and treatments related to intensive care unit-acquired muscle weakness: A cohort study. Aust Crit Care. 2021;34(5):435-445. doi:10.1016/j.aucc.2020.12.005

Eggmann S, Luder G, Verra ML, Irincheeva I, Bastiaenen CHG, Jakob SM. Functional ability and quality of life in critical illness survivors with intensive care unit acquired weakness: A secondary analysis of a randomised controlled trial. PLoS One. 2020;15(3):e0229725. Published 2020 Mar 4. doi:10.1371/journal.pone.0229725

Appleton RT, Kinsella J, Quasim T. The incidence of intensive care unit-acquired weakness syndromes: A systematic review. J Intensive Care Soc. 2015 May;16(2):126-136

Needham DM, Sepulveda KA, Dinglas VD, Chessare CM, Friedman LA, Bingham CO 3rd, et al. Core outcome measures for clinical research in acute respiratory failure survivors: an international modified Delphi consensus study. Am J Respir Crit Care Med. 2017;196(9):1122-1130. doi: 10.1164/rccm.201702-0372OC.

The tables are attached to the document "R1_Response to Reviewers".

-Minor comments:

line 104: EN is already defined in line 97.

Thank you for your correction. We have made the change.

-Lines 173-220: Write this section in simple, understandable text instead of listing items.

We have revised this section. We believe that it is now in a simpler and more understandable form to read.

-Discussion: Most of the information can be given by just adopting the numbered reference since the names of the authors appear repeatedly. Review the writing style at this level.

We have revised the wording of the discussion section. We have removed the names of the authors, leaving only the bibliographical reference.

---

## [Decision Letter · Decision Letter 1]

19 May 2023

Enteral nutrition management in critically ill adult patients and its relationship with intensive care unit-acquired muscle weakness: a national cohort study

PONE-D-23-04050R1

Dear Dr. Zaragoza-García,

We’re pleased to inform you that your manuscript has been judged scientifically suitable for publication and will be formally accepted for publication once it meets all outstanding technical requirements.

Kind regards,

Sebastien Kenmoe

Academic Editor

PLOS ONE

Additional Editor Comments (optional):

Reviewers' comments:

Reviewer's Responses to Questions

**Comments to the Author**

1. If the authors have adequately addressed your comments raised in a previous round of review and you feel that this manuscript is now acceptable for publication, you may indicate that here to bypass the “Comments to the Author” section, enter your conflict of interest statement in the “Confidential to Editor” section, and submit your "Accept" recommendation.

Reviewer #2: All comments have been addressed

2. Is the manuscript technically sound, and do the data support the conclusions?

Reviewer #2: Yes

3. Has the statistical analysis been performed appropriately and rigorously? 

Reviewer #2: N/A

4. Have the authors made all data underlying the findings in their manuscript fully available?

Reviewer #2: Yes

5. Is the manuscript presented in an intelligible fashion and written in standard English?

Reviewer #2: Yes

6. Review Comments to the Author

Reviewer #2: Authors have adequately addressed my comments raised in a previous round of review and I feel that this manuscript is now acceptable for publication. The major shadowed areas I mentioned have been resolved by the authors. The manuscript technically sound, and do the data with statistical analyses support the conclusions. The authors in their comments can make all data underlying the findings in their manuscript fully available. The current version of the manuscript is presented in an intelligible fashion and written in standard English.

7. PLOS authors have the option to publish the peer review history of their article (what does this mean?). If published, this will include your full peer review and any attached files.

Reviewer #2: No

---

## [Editor Report · Acceptance letter]

25 May 2023

PONE-D-23-04050R1 

Enteral nutrition management in critically ill adult patients and its relationship with intensive care unit-acquired muscle weakness: a national cohort study 

Dear Dr. Zaragoza-García:

I'm pleased to inform you that your manuscript has been deemed suitable for publication in PLOS ONE. Congratulations! Your manuscript is now with our production department. 

Kind regards, 

on behalf of

Dr. Sebastien Kenmoe 

Academic Editor

PLOS ONE